# Whole Body Substrate Metabolism during Different Exercise Intensities with Special Emphasis on Blood Protein Changes in Trained Subjects—A Pilot Study

**DOI:** 10.3390/jfmk8030102

**Published:** 2023-07-24

**Authors:** Wondyefraw Mekonen, Günther Schwaberger, Manfred Lamprecht, Peter Hofmann

**Affiliations:** 1Department of Physiology, College of Health Sciences, Tikur-Anbessa Medical School, Addis Ababa University, P.O. Box 5657, Addis Ababa 1165, Ethiopia; w.mekonen@gmail.com; 2Institute of Physiology & Pathophysiology, Medical University Graz, Neue Stiftingtalstraße 6, 8010 Graz, Austria; guenther.schwaberger@uni-graz.at; 3Institute of Nutrient Research, Petersbergenstrasse 95b, 8042 Graz, Austria; manfred.lamprecht@greenbeat.at; 4Division of Medicinal Chemistry, Neue Stiftingtalstrasse 6, 8010 Graz, Austria; 5Institute of Human Movement Science, Sport and Health, Exercise Physiology, Training Therapy Research Group, University of Graz, Aigner-Rollett-Allee 11, 8010 Graz, Austria

**Keywords:** exercise, amino acids, urea, FFA, glycerol, glucose

## Abstract

Contrary to carbohydrate and fat metabolism, the influence of a single exercise dose on protein metabolism has not been adequately explored yet. We assessed the effects of different exercise intensities and durations on blood protein changes and their association with carbohydrate (CHO) and fat metabolism in six eligible trained subjects. Subjects performed maximal incremental (IE_100_: at 100%VO_2max_) and submaximal continuous exercise (CE) at 75%VO_2max_ for 30 min (CE_75_) and at 50%VO_2max_ for 90 min (CE_50_). Blood samples were collected at rest (R), end of exercise (EE), and 1 h after recovery to assess blood urea nitrogen (BUN), plasma amino acids (AA), glucose, lactate, FFA, and glycerol. In IE_100_ blood lactate, CHO-oxidation (g/min), energy expenditure (kcal/min), and RER were significantly increased during rest (*p* < 0.05). CE_50_ induced significantly higher BUN, FFA, glycerol, and fat oxidation (g/min) (*p* < 0.05). At recovery, the mean sum of the free AA pool (µmol/L) reduced by 8% (*p* < 0.03) during CE_50_. Values for CE_75_ were between IE_100_ and CE_50_. Beside lipolysis, also proteolysis (BUN) was an important source of fuel for low-to-moderate intensity CE_50_. An increased uptake of AA from the plasma bed during CE_50_ suggests the importance for oxidation and synthesis of other metabolic sources such as gluconeogenesis necessary for recovery. Therefore, one needs to be cautious of protein diet following prolonged cycle exercise training.

## 1. Introduction

The role of carbohydrate (CHO) and fat as major sources of energy during physical exercise has been extensively demonstrated [1,2,3,4]. Consequently, CHO-oxidation supplies the major fuel during high exercise intensities (80–100%VO_2max_) of short durations, while fat oxidation supplies the major fuel during low-to-moderate exercise intensities (45–65%VO_2max_) of prolonged durations [4,5]. Information concerning the dose–response relationship of protein metabolism at different exercise intensities and/or duration, however, remains limited. Recently, Lee et al. [6] argued the existence of imbalances between dietary protein intake and dietary protein needs, which may result in net protein loss in athletes, leading to tissue protein breakdown to become a source of essential amino acids needed to maintain critical body functions. Although performance was not improved, protein supplement strategies were shown to reduce endurance exercise induced muscle damage [7]. Additionally, it was suggested that skeletal muscle mass loss with energy deficit may be due to protein breakdown by the autophagy-lysosome and the ubiquitin-proteasome systems and that exercise mitigated the loss of muscle mass by attenuating autophagy activation [8].

To evaluate protein metabolism during exercise, previous studies have used the production of blood urea nitrogen (BUN) as an index of protein catabolism [9,10,11]. It was shown that the flux of urea to the circulation was proportional to increased net-protein catabolism during endurance exercise [9,10,11,12]. This fact explains that the level of protein degradation during exercise can be evaluated from urea production because blood flow during exercise is diverted to the general circulation, and this effect enables urea to be easily extracted and evaluated. Other authors, however, do not entertain the use of urea as an index of protein catabolism [13,14]. It was argued that urea is excreted with urine or sweat during exercise, and thus, measuring urea during exercise may not lead to correct interpretations [14]. Independent of these discussions, a combination of biomarkers including total protein, albumin, globulin (calculated), blood urea nitrogen [BUN], nitrogen balance (calculated), and amino acid analysis was presented to help athletes gauge their protein status and make dietary alterations to improve training outcomes [6].

The mobilization of protein can also be observed from changes in total free plasma amino acid (TFAA) concentrations that occur during exercise and recovery periods [13,14]. Though TFAA level is small in concentration, i.e., only 2% of the total plasma amino acid pool [15], it is highly influenced by external stimuli like exercise intensity, duration, training status, diet, etc. [16,17]. Normally, amino acids of the plasma enter the liver or skeletal muscles quickly and are transformed into intermediates by other metabolic processes including gluconeogenesis, urea-genesis, and alanine-glucose cycle [16,17,18]. Thus, identifying the level of exercise intensity and/or duration that induces changes in TFAA may help us to realize which exercise load enhances the uptake of amino acids and their mobilizations during different exercise activities.

Our study also explored changes in groups of AAs including branched chain amino acids (BCAA) and sulfur-containing AAs during exercise and recovery periods. BCAA were explained to promote exercise capacity and lipid oxidation [19], though a recent review remarked inconsistent views, i.e., an increase, a decrease, or no change in the concentrations of BCAAs following different exercise bouts [20]. Works that dealt on sulfur-containing amino acids during endurance exercise remain scarce. Because high-intensity endurance bursts differ metabolically from prolonged continuous endurance training exercises [21], this study will report to which exercise intensities and/or durations these groups of AAs respond.

We, therefore, sought to compare the effect of three different exercises intensities including: maximal incremental (100%VO_2max_), two-continuous submaximal (75%VO_2max_, 30 min), and moderate (50%VO_2max_, 90 min) exercise intensities on changes in blood protein metabolites in active subjects. Markers for blood protein include blood urea nitrogen (BUN) and plasma amino acids. Because the body uses combinations of fuels to extract energy during exercise, markers for carbohydrate (glucose and lactate) and fat (FFA and glycerol) were assessed along with changes in protein metabolism during the different exercise bouts, and their associations with blood N2 sources were assessed. We hypothesized that a prolonged duration of moderate intensity exercise enhances the mobilization of blood N2 sources compared to heavier intensities of shorter durations.

## 2. Materials and Methods

### 2.1. Subjects

Nine endurance-trained male subjects with mean ± SD, age 29 ± 5 years, height 181 ± 3 cm, weight 76 ± 4 kg, BMI 23 ± 0, and VO_2max_ 63.9 ± 1 mL/kg/min participated in the study. Three subjects with a VO_2peak_ < 50 mL/kg/min who engaged in physical training for <2 h/wks were considered not sufficiently trained and were not included in the study. Six subjects fulfilled the criterion and provided a written consent. The ethical committee of a local university approved the protocol (GZ. 39/158). Subjects were instructed to maintain their regular diet, which consisted of 65% CHO, 21% fat, and 14% protein. A dietician controlled the diet and estimated the caloric cost to be around 3015 kcal/day. Subjects were informed to refrain from physical activity 24 h before the start of the tests, and each subject served as his own control.

### 2.2. Exercise Protocol

All exercise tests were performed in the morning hours between 8–9 am in order to avoid possible influences of circadian variation. Each subject rested for about 10 min before the commencement of the test. Room temperature was maintained at about 20 °C with an air conditioner, and a ventilator was allowed to ensure proper circulation. A butterfly catheter was inserted into an antecubital forearm vein to draw blood when needed. Catheter patency was maintained by intermittent flushing with 0.9% physiological saline. For basal rest measurement, 5 mL of venous blood was withdrawn and transferred into a tube with EDTA. Before each test, the gas analyzer was calibrated with gases of known concentration (16%O_2_ and 5%CO_2_).

The first test included the determination of maximal exercise (100%VO_2max_) by a progressive incremental test on a cycle ergometer (Monarch, Vansbro, Sweden). This maximal test started with a 3 min warm-up period at 40 Watt (W), and the workload was gradually increased by 20 W/min until the subject terminated the test due to voluntary exhaustion. Oxygen uptake (VO_2_), carbon dioxide output (VCO_2_), and ventilation (VE) were measured continuously using a breath-by-breath online system (Oxycon Pro, Jaeger, Wurzburg, Germany). The criterion to achieve the VO_2max_ was RER > 1.1, an increase in VO_2_ of <2 m/kg/min over the previous work rate, and the inability to maintain the pedaling frequency. Heart rate was monitored continuously (Sport-Tester PE4000, Polar Electro, Kempele, Finland). From peak O_2_ consumption (100%VO_2max_), workloads corresponding to submaximal intensity (75%VO_2max_, 250 W, 30 min) and a moderate intensity (50%VO_2max_, 150 W, 90 min) were calculated, and participants performed these tests on 5-day intervals.

### 2.3. Measurements

Energy Expenditure (EE): EE in (kcal) during the three exercise intensities was calculated from VO_2_ (L/min) and RQ according to Weir’s predictive equation [22].

CHO and Fat Oxidation: These were computed in (g/min) from VO_2_ (L/min) and CO_2_ (L/min) using the stoichiometric equation of Peronnet and Massicottee, 1991 [23].

Blood sampling and analysis: A 5 mL volume of venous blood samples were collected at the end of each exercise duration and after 1 h of recovery and were transferred into vacutainer tubes containing dry disodium EDTA (1 mg/mL). Plasma amino acids were determined using an automatic amino acid analyzer (Biotronic 7000, Munich, Germany), which uses ion-exchange column chromatography. Data were analyzed at the Biochemistry Department of the local University. Blood urea was determined enzymatically using an auto-analyzer (Technicon Instruments, Tarrytown, NY, USA). FFA was determined usng test kits (Wako, Neuss, Germany). Lactate and blood glucose concentrations were determined from earlobe capillary blood using an enzymatic method (Eppendorf Ebio+, Wesseling-Berzdorf, Germany). Test kits (Boehringer Mannheim, Mannheim, Germany) were used to analyze glycerol concentrations enzymatically.

### 2.4. Statistic Analysis

Data were presented as mean ± SD, and SPSS statistical package (ver.20) was used for all statistical calculations. Nonparametric Friedman’s test was applied, and when the critical F-test showed significance, Wilcoxon’s post hoc comparisons were used to see differences between two mean values of different variables by considering *p* < 0.05 as a significant difference.

## 3. Results

### 3.1. Cardio-Respiratory Changes

Oxygen uptake (L/min) and heart rate (b/min) at the end of exercise were significantly increased from rest (*p* < 0.05) both during the maximal and submaximal exercise intensities, but not for moderate intensity (Table 1). However, O_2_ uptake in the maximal incremental test was 22% higher (4.8 vs. 3.7 L/min) compared to the submaximal intensity. RER was the highest during the maximal test at 1.20 ± 0.05 and the lowest at 0.87 ± 0.05 during the moderate intensity test of 90 min duration. In between the two tests (i.e., during 30 min exercise), RER showed a value of 0.98 ± 0.03, reflecting the expected intensity-dependent differences in carbohydrate and fat metabolisms (Table 1).

### 3.2. Energy Expenditure (EE in kcal)

Mean ± SD value for EE at the end of the maximal incremental (IE_100_) intensity (15 min) and continuous submaximal (CE_75_, 30 min) and moderate (CE_50_, 90 min) intensities were all significantly higher (*p* < 0.03) compared to basal rest.

### 3.3. Changes in CHO Metabolites

#### 3.3.1. Lactate

As seen in Table 1, blood lactate concentration at the end of the maximal incremental test significantly increased by ten-fold from rest (*p* < 0.05). At the end of the submaximal test, blood lactate was significantly raised by seven-fold from rest. Comparing mean differences between the two-exercise tests showed that blood lactate was lower by 34% during the submaximal than the maximal exercise test. At the end of the moderate exercise test (50%VO_2max_, 90 min), blood lactate remained unchanged compared with rest (*p* > 0.05).

#### 3.3.2. Glucose

Blood glucose concentration significantly decreased from rest (*p* < 0.05) both during the maximal and the moderate exercise intensities, but remained unchanged during the submaximal (30 min) exercise tests (Table 2). At recovery, blood glucose concentration was not significantly different from rest in all three exercise bouts.

#### 3.3.3. Carbohydrate (CHO) Oxidation

CHO-oxidation was the highest during 15 min of maximal exercise (12.38 ± 4.33 g/min) and lowest during moderate intensity of 90 min exercise (1.68 ± 0.42 g/min) compared to rest (0.47 ± 0.25). CHO oxidation at the end of the submaximal intensity test was (4.67 ± 0.93 g/min), which is between the maximal and the moderate exercise intensity. CHO oxidation during the different intensities/durations is shown in Table 1.

### 3.4. Changes in Fat Metabolism

#### 3.4.1. FFA

Both at the end of exercise and the recovery period of moderate intensity (90 min), blood FFA was significantly increased by about three-fold (0.3 ± 0.01 vs. 1.1 ± 0.1, *p* < 0.03) compared to rest (Table 2). Blood FFA concentration was unchanged at maximal (IE100) and submaximal (CE_75_, 30 min) exercise intensities of shorter durations (Table 2).

#### 3.4.2. Glycerol

Both during submaximal (30 min) and moderate (90 min) intensities and their respective recovery periods, blood glycerol concentration was significantly increased (*p* < 0.03) compared to basal rest (Table 2). The increase in glycerol level during the 30 min submaximal exercise was three-fold, while that of 90 min moderate exercise was five-fold compared to their baseline levels. Contrary, the short duration (15 min) incremental maximal exercise did not induce a significant change (*p* > 0.05) in blood glycerol concentration compared with basal rest level (Table 2).

#### 3.4.3. Fat Oxidation

Fat oxidation was lowest (0.04 ± 0.09 vs. −1.43 ± 0.50 g/min) during the maximal incremental test (15 min) compared with rest level, a reduction of about 36 times compared to rest (Table 1). Contrarily, fat oxidation was the highest during 90 min of the moderate intensity compared to rest (Table 1). Fat oxidation at 30 min submaximal intensity was 0.09 ± 0.24 g/min, a value between that of maximal and moderate exercise intensities.

### 3.5. Changes in Blood Protein Sources

#### 3.5.1. Blood Urea Nitrogen (BUN)

Blood urea concentration significantly increased from rest during moderate intensity of prolonged duration (90 min), and this significance persisted throughout recovery (Table 2). Blood urea remained unchanged (*p* > 0.05) during the heavier loads of maximal incremental and submaximal continuous exercise intensities (Table 2).

#### 3.5.2. Plasma Amino Acids (AAs)

Changes in plasma amino acids recorded at basal rest, end of exercise, and post-exercise recovery is presented in (Table 3). The maximal incremental exercise induced significant increases in plasma alanine, glutamine, citrulline, arginine, and tryptophan levels. However, when corrected for plasma volume changes, these significant differences disappeared. All other plasma amino acids remained unchanged when compared with their corresponding basal rest levels (Table 3). The mean total sum of plasma amino acid (TSPAA) level of maximal and submaximal intensities were higher than their respective basal rest levels (e.g., maximal: rest = 1698 vs. EE = 1894 vs. PE = 1781 µmol/L) (Table 3). However, the mean TSPAA level of prolonged (90 min) exercise showed a progressive decrease of about 8% at recovery compared with basal rest (e.g., rest = 1639 vs. EE = 1642 vs. PE = 1513 in µmol/L) (Table 3).

#### 3.5.3. BCAAs (Branched Chain Amino Acids)

BCAAs, which include valine, leucine, and isoleucine, did not show significant differences both during exercise and recovery periods in all of the three exercise intensities. Similarly, no significant changes were observed from basal rest on sulfur-containing amino acids, namely, methionine and tryptophan (Table 3).

## 4. Discussion

The finding of this study is that there is a significant increase in blood urea concentration during prolonged exercise (CE_50_) compared to higher intensities and is an indication of a higher rate of protein catabolism having occurred in addition to the well-known fact of higher fat oxidation [1,2,3,4,5,6] (Table 2). We can thus suggest from our study that not only lipolysis but also proteolysis remains an important source of fuel for prolonged exercise activities that may partially supply the energy needs for this type of exercise activity. Our work is in line with a recent article by Liang et al. [7] that showed reduced muscle damage induced by endurance exercise with protein and carbohydrate supplementation but no improvement in performance.

### 4.1. Maximal Incremental Intensity (IE_100_: At 100%VO_2max_)

The significantly increased blood lactate concentration, higher RER > 1.2, and an increased rate of CHO-oxidation exclusively indicate that CHO-oxidation was the main source of fuel during the short duration (<15 min) of the maximal (IE_100_) intensity exercise. These results are in line with several previous works that showed an increased rate of CHO-oxidation (glycolysis) that usually occurs during heavy exercises above the second lactate and/or ventilatory thresholds [17,24]. As it is known, the accumulation of lactate in the blood is positively correlated with the production of acid in the blood plasma [18]. Acidosis normally decreases Ca^2+^ binding capacity of muscle fibers and continuously impedes muscle contraction and their motility [18,25]. Thus, as results of this study indicate, the significantly increased blood lactate level (by almost 10-fold) may have forced our subjects to quit exercising within a short duration of less than 15 min because of the higher production of acids in their blood plasma (Table 2).

Blood N_2_ sources, blood urea nitrogen (BUN), and most of plasma amino acids remained unchanged compared to rest (*p* > 0.05) both during the maximal exercise (IE_100_) and the recovery period (Table 3). In addition, groups of amino acids including BCAA (valine, leucine, and isoleucine) and sulfur-containing amino acids, e.g., methionine (met), did not change significantly from basal rest during this exercise test (Table 3). This lack of significant change in blood urea and plasma amino acids may be explained by the increased rate of CHO-oxidation that was responsible for the higher rate of acid production. We suggest that acidosis that decreases pH [18] may have also inhibited the activity of blood protein pathways from being utilized during such an exhausting maximal (IE_100_) exercise test. Similar to our study, previous works have observed no changes in protein metabolism during exercise and recovery periods of heavy exercise activities [21,26,27].

In this study, markers of fat oxidation including blood FFA and glycerol concentrations did not change from rest (*p* > 0.05) during the exercise and recovery period of the maximal (IE_100_) exercise, suggesting that fat oxidation was not a major source of fuel for such a short duration of high-intensity exercises. To justify this metabolic effect, the rate of fat oxidation calculated in g/min (Table 2) in this study reduced significantly by 35% from rest (0.04 ± 0.09 vs. −1.43 ± 0.50, *p*< 0.03) (Table 1). The summation of these three results (i.e., a decrease in blood FFA, glycerol levels, and a decrease in fat oxidation) generally support our and previous views that [1,4,5,6,21] fat oxidation was not a major source of fuel for high exercise intensities of short duration in trained subjects. As it was explained, the inhibition of lipolysis during heavy exercise occurs due to low pH [28], or possibly, due to diminished blood flow to adipose tissues that hinders the disposal of FFA to the working muscles [29]. Independent of these causes, we now suggest that the limitation of protein degradation (i.e., proteolysis) also plays a negligible role in exercise energy metabolism during heavy short-duration exercises above the second lactate and/or ventilatory threshold level.

### 4.2. Submaximal Continuous Exercise Intensity (CE_75_: At 75%VO_2max_), 30 min

During the continuous submaximal exercise, energy provisions in the form of glycolysis (CHO-oxidation) and lipolysis (fat oxidation) persisted throughout the exercise and recovery periods. In our study, these dual metabolic adjustments were demonstrated by significant increases both in blood lactate (by eight-fold) and glycerol concentrations (by three fold) in blood plasma (Table 2). It is important to note that an increase in blood glycerol concentration reflects a higher degree of lipolysis because skeletal muscles and adipose tissues lack the enzyme glycerol kinase to uptake and utilize glycerol [30]. Although FFA release into the blood also indicates lipolysis, FFAs can be re-esterified back to TG in adipose tissues [30]. Therefore, compared to glycerol, FFAs may be a limited indicator of the degree of lipolysis during exercise. In our study, the three-fold significant increase in glycerol level justified the presence of lipolysis during the submaximal exercise bout despite no changes in FFAs [31]. Blood lactate concentration during the submaximal load in this study was, however, lower by 34% compared to the maximal (IE_100_) exercise, showing a lower degree of CHO-oxidation during the submaximal (CE_75_) exercise. Moreover, RER at the end of the submaximal load was below 1.0 (Table 2), indicating that it is below the second threshold intensity [32]. We, therefore, suggest from our results of dual metabolic observations that both CHO and fat oxidations have contributed to a more-or-less balanced energy provision during the 30 min of submaximal exercise intensity (CE_75_).

Concerning blood N_2_ sources, no significant changes in blood urea (Table 2) and plasma amino acid concentrations (Table 3) were observed compared to basal rest during the exercise and recovery period of the submaximal continuous (CE_75_) intensity. In addition, the total sum of plasma amino acid (TSPAA) concentration (Table 3) remained unchanged during this intensity, indicating that the submaximal load was not long enough to stimulate a significant change on blood N_2_ sources. Physiologically speaking, there is obviously no need for proteins to be catabolized for fuel during heavier exercise intensities at least for the relatively short-duration exercises. Recently it was shown that the duration of exercise needs to be treated as an individual performance variable [32] and that a fixed duration does not reflect the individual maximal durability of subjects. Additionally, it was shown that subjects were able to cycle for 69.6 ± 14.8 min (range: 40–90 min) at a comparable intensity of 73.3 ± 4.8% VO_2max_, which shows that the limits of duration have not been reached in our study in the submaximal exercise test. Additional studies need to be performed applying maximal duration as several physiological variables were shown to respond when approaching the end of exercise [32].

### 4.3. Moderate Prolonged Continuous Exercise Intensity (CE_50_: 50%VO_2max_), 90 min

Because of the low-to-moderate exercise intensity and corresponding low near resting values of blood lactate concentration (Table 1), our participants were able to exercise for a relatively long period of 90 min duration without showing exhaustion. The ability to exercise for longer periods during such prolonged exercise loads was described previously by the study of Moser et al. [33]. As an indicator of the low intensity, RER was the lowest (Table 1) and fat oxidation was the highest (0.06 ± 0.07 vs. 0.40 ± 0.29 g/min, *p* < 0.014) during this exercise intensity (Table 2).

Concerning blood N_2_ sources, blood urea, the product of protein catabolism, showed a 12% significant increase from rest (*p* < 0.03), suggesting a higher rate of protein breakdown that occurred during this exercise intensity (Table 2). In support of this, Lemon et al. [34] explained that an increase in protein catabolism occurs during prolonged exercise activities because of diminished energy supply from carbohydrate sources. In line with this, blood glucose and endogenous carbohydrate stores were shown to decrease during continuous endurance exercises longer than 90 min moderate intensity exercises of 30–70% of one’s VO_2max_ level [35]. In our study, the significantly decreased blood glucose level from rest (91.3 ± 6.1 vs. 81.3 ± 2.2, *p* < 0.05) and the lowest rate of CHO-oxidation calculated during the prolonged exercise support the view of lower energy supply from CHO sources and the requirements of other blood substrates (e.g., proteins or lipids) to replace as fuel sources.

Apart from changes in blood urea concentration, we have also calculated differences that can be observed on the plasma mean total sum of plasma amino acid (TSPAA) pool level that may occur at basal rest, end of exercise, and the recovery period of each exercise intensity (Table 3). A significant reduction of about 8% from rest (i.e., 1639 vs. 1513 µmol/L) was calculated for the mean TSPAA pool during the recovery period of the continuous prolonged exercise (CE_50_). The drastic reduction in amino acids from the plasma bed may indicate the use of amino acids for the synthesis or production of intermediates of other metabolic processes (e.g., gluconeogenesis, urea genesis, and tricarboxylic acid cycle) that usually occurs in the liver or skeletal muscles [36]. A previous work on amino acid metabolism during endurance exercise [36] explained that amino acids utilized during recovery could be used for glycogen restitution, which is necessary to maintain blood glucose levels [36,37]. In comparison, heavier intensities of short duration (>75%VO_2max_, <30 min) did not show decreases in the plasma TSPAA level, implying that other blood substrates were more important than the catabolism of blood N_2_ sources in supplying fuel during heavy exercise loads. Realizing that endurance exercise of prolonged duration caused depletion of glycogen stores, the uptake of amino acids from the plasma bed, which was also reflected by a significant rise in blood urea concentration, indicated the higher rate of protein degradation that was necessary to augment the energy needs during the continuous prolonged cycle exercise tests.

In our study, the three workloads, i.e., maximal incremental (IE_100_), submaximal (CE_75_), and moderate (CE_50_), did not induce significant changes on BCAAs, suggesting that exercise intensity in combination with the applied duration (overall load) had little effect on changes of BCAAs. Similarly, sulfur-containing amino acids also did not show significant changes from rest following the three different exercise intensities. The heterogeneity of these results may be expected from different intensities and durations as well as the fact that even a similar fixed duration may result in different individual responses due to individual durability [32]. A recent review [20] on changes in BCAAs also explained the lack of change, a decrease, or an increase in levels following endurance exercise activities [20].

As shown in several studies [1,2,3,4,5], prolonged exercise (90 min) of moderate intensity (CE_50_) resulted in a higher rate of fat oxidation (lipolysis) compared to heavier intensities of short duration. The higher rate of fat oxidation (lipolysis) demonstrated a significant four-fold increase in FFA and a six-fold increase in glycerol concentrations, which persisted throughout recovery periods (Table 2). These results further justify the importance of fat oxidation as a major source of fuel for moderate exercise intensity of prolonged duration.

### 4.4. Limitations

This study included only a small group of male subjects, and therefore, results may not be representative of all sexes and performance levels. Thus, metabolic changes that may occur in females of the same age and activity need to be addressed in order to support the findings of this study as well as studies in untrained and highly trained athletes. Additionally, future work needs to include more sophisticated approaches to determine muscle protein status in relation to long- and medium-duration exercise bouts to better understand mechanistically the sources of amino acids used in substrate turnover, energy flux, and replenishment of TCA cycle intermediates. However, there is still a lack of studies on protein metabolism during exercise, and our study may serve as a study design model for future investigations not only in trained subjects but much more in the general public and especially the overweight population.

## 5. Conclusions

The provision of energy during prolonged exercises (90 min) of moderate intensity (CE_50_) is enhanced during both fat oxidation (lipolysis) and protein degradation (proteolysis) in exercising subjects. In heavier exercises (>75%VO_2max_, <30 min) of short durations, CHO-oxidation is the major source of fuel without a significant contribution from protein and fat oxidations. Athletes and coaches should be cautious of their protein diet following prolonged cycle exercise training.

## Figures and Tables

**Table 1 jfmk-08-00102-t001:** Cardio-respiratory and blood lactate (mmol/L) concentrations measured during the different exercise intensities and/or durations plus recovery period.

	Maximal Load (100% VO_2max_)		Submaximal (75% VO_2max_)		Moderate (50% VO_2max_)	
	Rest (0)	15 min	Rest (0)	30 min	Rest (0)	90 min
VO_2max_ (L/min)	0.3 ± 0.1	4.8 ± 0.4 *	0.3 ± 0.01	3.7 ± 0.03 *	0.3 ± 0.0	2.3 ± 0.3
HR (b/min)	71.2 ± 4.5	190.7 ± 4.1 *	69.3 ± 5.4	182.3 ± 3.5 *	68.2 ± 5.3	132.5 ± 5.3 *
RER (--)	0.99 ± 0.1	1.20 ± 0.0	0.94 ± 0.02	0.98 ± 0.01	0.88 ± 0.0	0.89 ± 0.03
Lactate (mmol/L)	1.0 ± 0.2	10.3 ± 1.1 *	0.8 ± 0.1	6.8 ± 1.3 *	0.8 ± 0.1	0.8 ± 0.20
CHO-Oxo (g/min)	0.35 ± 0.14	12.38 ± 4.33 *	0.37 ± 0.11	4.67 ± 0.93 *	0.24 ± 0.18	1.68 ± 0.42 *
Fat-Oxo (g/min)	0.04 ± 0.09	−1.43 ± 0.50 *	0.12 ± 0.19	0.09 ± 0.24	0.06 ± 0.07	0.40 ± 0.29 *

*, *p* < 0.05 all values compared with the corresponding rest period. CHO-oxo: carbohydrate oxidation.

**Table 2 jfmk-08-00102-t002:** Changes in blood substrate concentrations measured during the three different exercise intensities and/or durations plus recovery period (mean ± SD).

	Intensity and Duration	Basal Rest	End of Exercise	Recovery (1 h)
Glucose (mg/dL)	Max-incremental (15 min)	110.0 ± 6.8	103.3 ± 6.8 *	112.3 ± 12.7
	Submaximal (30 min)	100.5 ± 8.2	99.5 ± 9.7	96.3 ± 8.6
	Moderate (90 min)	91.3 ± 6.1	81.3 ± 2.2 *	88.7 ± 3.1
FFA (µmol/L)	Max-incremental (15 min)	0.3 ± 0.0	0.3 ± 0.1	0.5 ± 0.3
	Submaximal (30 min)	0.3 ± 0.2	0.4 ± 0.1	0.4 ± 0.2
	Moderate (90 min)	0.3 ± 0.1	1.1 ± 0.1 *	0.8 ± 0.1 *
Glycerol (µmol/L)	Max-incremental (15 min)	0.06 ± 0.01	0.07 ± 0.01	0.08 ± 0.02
	Submaximal (30 min)	0.06 ± 0.01	0.17 ± 0.04 *	0.07 ± 0.01
	Moderate (90 min)	0.05 ± 0.01	0.30 ± 0.04 *	0.12 ± 0.02 *
Urea (mg/dL)	Max-incremental (15 min)	31 ± 20	31 ± 4	33 ± 3
	Submaximal (30 min)	33 ± 40	33 ± 3	34 ± 3
	Moderate (90 min)	33 ± 40	37 ± 4 *	37 ± 3 *
TP (mg/dL)	Max-incremental (15 min)	6.67 ± 0.07	7.30 ± 0.16 *	6.75 ± 0.06 *
	Submaximal (30 min)	6.71 ± 0.09	7.13 ± 0.06 *	6.77± 0.10
	Moderate (90 min)	6.69 ± 0.04	7.03 ± 0.09 *	6.78 ± 0.03
%PV Change (%)	Max-incremental (15 min)	-	−9.49 ± 1.96	−1.29 ± 1.14
	Submaximal (30 min)	-	−6.28 ± 1.35	−0.88 ± 1.15
	Moderate (90 min)	-	−5.10 ± 1.01	−1.87± 0.79

*, *p* < 0.05 all value were compared to the initial basal rest level; %PV, shows percent (%) change in plasma volume from basal rest during each exercise.

**Table 3 jfmk-08-00102-t003:** Concentrations of plasma amino acids (µmol/L), TSAA (total sum of amino acids), and BCAA (branched chain amino acids) measured during three exercise intensities and the recovery period.

	Maximal (100% VO_2max_) Duration (X ± SE) = 15 min			Submaximal (75% VO_2_max) Duration (X ± SE) = 30 min			Moderate (50% VO_2max_) Duration (X ± SE) = 90 min		
(µmol/L)	Rest	15 min	PE-1 h	Rest	30 min	PE-1 h	Rest	90 min	PE-1 h
Val	214 ± 13	225 ± 13	228 ± 23	233 ± 24	229 ± 15	247 ± 18	228 ± 21	221 ± 24	211 ± 23
Leu	125 ± 8	137 ± 11	130 ± 18	123 ± 27	120 ± 20	125 ± 10	123 ± 13	133 ± 17	126 ± 15
Ile	65 ± 5	72 ± 6	76 ± 10	85 ± 14	83 ± 6	89 ± 8	76 ± 10	74 ± 11	72 ± 11
Gln	505 ± 29	563 ± 32 *	529 ± 23	502 ± 26	527 ± 25	490 ± 36	493 ± 43	482 ± 46	455 ± 42
Ala	373 ± 67	471 ± 91 *	428 ± 76	455 ± 36	568 ± 84 *	427 ± 59	421 ± 33	366 ± 55	303 ± 44
Pala	61 ± 9	61 ± 8	56 ± 6	56 ± 4	59 ± 4	60 ± 3	56 ± 4	55 ± 3	49 ± 3
Tyr	64 ± 9	66 ± 8	60 ± 5	64 ± 5	67 ± 4	70 ± 3	63 ± 5	66 ± 6	59 ± 4
Met	28 ± 5	29 ± 7	26 ± 5	27 ± 3	27 ± 5	27 ± 8	24 ± 4	23 ± 3	22 ± 2
Orn	56 ± 6	56 ± 3	51 ± 4	60 ± 3	53 ± 1	61 ± 3	56 ± 6	52 ± 3	50 ± 4
Cit	38 ± 6	43 ± 7 *	39 ± 7	33 ± 4	35 ± 3	36 ± 5	28 ± 3	37 ± 5 *	28 ± 3
Asp	12 ± 1	16 ± 1	14 ± 1	13 ± 1	15 ± 1	14 ± 1	13 ± 1	13 ± 2	11 ± 1
Arg	92 ± 9	106 ± 11 *	91 ± 8	72 ± 12	93 ± 7	101 ± 8	85 ± 15	70 ± 12	78 ± 10
3-MH	4 ± 1	5 ± 1	4 ± 1	5 ± 0	6 ± 0	6 ± 1	5 ± 1	6 ± 1	5 ± 1
Trp	52 ± 6	44 ± 5 *	49 ± 5	53 ± 5	50 ± 5	56 ± 3	48 ± 3	44 ± 4	44 ± 2
TSFAA ^$^	1689	1894	1781	1781	1932	1836	1639	1642	1513
BCAA ^and^	405 ± 29	432 ± 30	434 ± 52	441 ± 65	432 ± 41	461 ± 36	427 ± 39	428 ± 52	409 ± 49

*, *p* < 0.05 compared to the respective rest level in each of the exercise intensity (Wilcoxon test); ^$^, TSFAA (indicates mean total sum of free amino acid levels in µmol/L); ^and^, BCAA, indicates branched chain amino acids (leucine, isoleucine, and valine) in µmol/L. Note: AA level at rest, end of exercise, and at recovery of the prolonged exercise (1689 vs. 1513), a decrease of about 10%.

## Data Availability

The data presented in this study are available on request from the corresponding author. The data are not publicly available due to hospital confidentiality.

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
