# Peer review of "Whole Body Substrate Metabolism during Different Exercise Intensities with Special Emphasis on Blood Protein Changes in Trained Subjects—A Pilot Study"

_jfmk, 2023, doi:10.3390/jfmk8030102_

Round 1

Reviewer 1 Report

Thank you for the opportunity to review this paper. It wanted to assess the influence of exercise intentity and duration on blood protein changes and the association with carbohydrate and fat metabolism. The manuscript presents several major limitations limiting its use and quality.

First, it included only 6 participants, despite the study typology and the analysis performed, the number is too low. Please, justify it with a power analysis.

The second limitation is related to the term adopted “trained subject”, please, find a citation that confirm the >2 hours a week is enough to determine an athlete trained. 

The third and most important limitation is related to the training. It has been studied the acute effect of different training modalities.

Author Response

We thank the reviewer for the valuable comments and suggestions and we adapted the text accordingly. The response can be seen in the attached file 

Reviewer 2 Report

This paper examines the effects of three different cycling exercise protocols on substrate utilization in highly trained male endurance athletes. Specific comments are below:

1.       The notion that dietary protein can offset or prevent muscle protein loss is often repeated as a rationale for high doses and frequency of protein feeding and supplement use in athletics. However, studies from long term fasting and time restricted feeding (Longo, Mattson, and others) have added nuance and context to such assertions. It is an unjustified leap of logic to conclude that proteolysis must necessarily refer to skeletal muscle protein degradation. Indeed, protein recycling into individual amino acids via autophagy has been associated with enhanced metabolic health, insulin sensitivity, and glucose homeostasis. Thus this reviewer indicates that a bit more context is warranted when discussing proteolysis pertaining to diet and exercise because protein catabolism and skeletal muscle catabolism are often improperly conflated in discussions of substrate utilization.

2.       The organization of the article is excellent and the logical flow is very easy to follow. Readability and clarity is very good throughout. The structure of the discussion is very well planned and this section is well-written. The authors are to be commended for this.

3.       The emphasis on duration as an individual performance variable is noteworthy and has a degree of novelty in the context of this paper. The analysis of the metabolic parameters measured in the 75% intensity exercise experiment are a valuable addition to the knowledge base and would be of great interest to scientists aiming to clarify substrate utilization in trained endurance athletes at intensities in this range since they are correlated with maximal sustainable power output during competition. This is a major determining factor in results in endurance events.

4.       The section on long duration, low intensity cycling is carefully written and the authors are mindful of context when discussing the contributions of protein catabolism to energy flux and metabolic homeostasis. Blood urea nitrogen is a rather coarse indicator of whole body protein catabolism and cannot provide any deterministic information about the source of amino acids that were degraded and processed for excretion by the urea cycle. In light of the increasing popularity of time restricted feeding as a modality for increasing metabolic health, largely postulated to result from autophagy, precision is needed when discussing protein catabolism. Protein catabolism is often exaggerated as a justification for excessive caloric and protein intake and is used by commercial interests in the supplement industry to sell products. This study could be extended in future work by using more sophisticated approaches to determine muscle protein status in response to long and medium duration exercise bouts over a chronic period to better understand the exact sources of amino acids used in substrate turnover, energy flux, and the replenishment of TCA cycle intermediates. The authors should also emphasize that during low intensity, long duration endurance exercise circulating substrates are sufficient to supply the ATP-flux necessary to maintain power output. The high rates of lipolysis, as evidenced by increased glycerol concentrations, discussed by the authors are a testament to the abundant use of lipids as the main energy source for such exercise bouts and logically are a counterpoint to the notion that excess protein catabolism should be of inordinate concern when considering dietary practices aimed at supporting athletic performance. While highly trained endurance athletes are hardly ever overweight, the general public should not be neglected when discussing diet in the context of exercise training for a general audience seeking mainly to use exercise as part of an overall strategy for health maintenance and promotion.

Author Response

We thank the reviewer for the valuable comments and suggestions and adapted the text accordingly. The response can be seen in the attached file.

Reviewer 3 Report

The manuscript "Whole body substrate metabolism during different exercise intensities with special emphasis on blood protein changes in trained subjects" is potentially interesting and aims to evaluate the contribution and potential of amino acid oxidation in three different exercise protocols.

The manuscript presents several critical issues that do not allow it to be accepted in its current form.

Here are some comments:

1) It is now known that short-term/high-intensity exercises use the catabolism of carbohydrates and then marginally that of lipids and, albeit less so, that of amino acids.

Wouldn't it have been more useful to use three different moderate-intensity aerobic protocols in, for example, those that mimic three different types of sporting activity?

2) Remodulate the abstract highlighting the rationale and the objective of the work.

3) Lines 94-95. The study was approved by the scientific ethics committee. Please provide approval code references.

4) Only 6 subjects participated in the study. So it's a preliminary study. Review the manuscript accordingly.

5) What is the statistical power of the study?

6) Lines 130-132. Please, given the importance of the data, describe in detail the method applied with the Biotronic7000, München, Germany.

7) Lines 295-297. Please explain this phenomenon better since the level of fatty acids does not increase. (In moderate exercise instead increasing the use of lipids increase both parameters).

8) AA level at rest, end of exercise, and at recovery of the prolonged exercise (1689 vs. 1513), a decrease by about 10%.

It is not clear why to refer to the value of 1689 and not to the average between 1689, 1639 and 1781.

Also, did these 6 subjects progressively perform the three types of exercise?

Are they different subjects? It is not clear.

Please, some parts of the manuscript should be reviewed by a native-English speaker author.

Author Response

(The authors gave the same response as above.)

Reviewer 4 Report

Dear Authors,

in my opinion, the content of the manuscript requires minor corrections. The quality of this article could be improved with the suggestions below.

1.      Both in the title of the manuscript, as well as in the "Material and methods" section, it is worth supplementing the information on the nature of the research conducted - "case series".

2.      Notes on the "Material and Methods" section:

If the studies were registered prospectively or retrospectively, please provide the register number.

3.      In the "Results" section: table 2, in the "end of exercise" column, it is worth adding information about how much time had passed from the end of the exercise until the moment when measurements were made (lines: 164).

4.      In the "Limitations" section, the Authors should focus only on the limitations of their own research. The comment on the directions of further research should be removed (lines: 384-386).

5.      Minor punctuation errors should be corrected throughout the text.

 Does not require major corrections.

Author Response

(The authors gave the same response as above.)

Round 2

Reviewer 1 Report

I don't think the quality of this manuscript is sufficient to be published.

Author Response

Dear reviewer,

We thank for the suggestions and agree that a larger study is necessary to evaluate the pilot data obtained in this study. However, we are convinced that these pilot data are relevant to the field and new as such an approach has rarely been published. 

Reviewer 3 Report

I thank the authors for modifying the manuscript in accordance with the critical issues identified.

Some points still need to be clarified:

a- "Plasma amino acids were determined by an automatic amino acid analyzer, which uses ion-exchange column chromatography (Biotronic 7000, Munich, Germany). Data were analyzed at the biochemistry department by a laboratory technician specialized in these analyses."

WHICH LAB? WHICH UNIVERSITY?

b-7) Lines 295-297. Please explain this phenomenon better since the level of fatty acids does not increase. (In moderate exercise instead increasing the use of lipids increase both parameters). Response: Thanks for the comment. The main text has been adapted accordingly.

WHERE IN THE MANUSCRIPT IS THE REVIEW? WHICH COMMENT WAS ADDED?

Please check again the manuscript.

Author Response

Dear reviewer,

we thank for the valuable suggestions and remarks. We addressed all issues raised and we adapted the text accordingly including an additional reference.

The main text has been re-read and some minor changes have been performed to improve writing and style.
